



# Modeling simulation of aerosol light absorption over the Beijing-Tianjin-Hebei region: the impact of mixing state and aging processes

Huiyun Du[1], Jie Li[1,2*], Xueshun Chen[1*], Gabriele Curci[3], Fangqun Yu[4], Yele Sun[1], Xu Dao[5], Song Guo[6], Zhe Wang[1], Wenyi Yang[7], Lianfang Wei[1], Zifa Wang[1,2]

[1]State Key Laboratory of Atmospheric Boundary Layer Physics and Atmospheric Chemistry (LAPC), Institute of Atmospheric Physics, Chinese Academy of Sciences, Beijing 100029, China
[2]College of Earth and Planetary Sciences, University of Chinese Academy of Sciences, Beijing 100049, China
[3]Department of Physical and Chemical Sciences, University of L'Aquila, L'Aquila, Italy
[4]Atmospheric Sciences Research Centre, University at Albany, Albany, NY, USA
[5]China National Environmental Monitoring Centre, Beijing 100012, China
[6]College of Environmental Sciences and Engineering, Peking University, Beijing 100871, China
[7]Chinese Academy of Environmental Planning

Correspondence to: Jie Li (lijie8074@mail.iap.ac.cn), Xueshun Chen (chenxsh@mail.iap.ac.cn)

**Abstract.** The mixing state and aging characteristics of black carbon (BC) aerosols are the key factors in calculating their optical properties and quantifying their impacts on radiation balance and global climate change. Considerable uncertainty still exists in the absorption properties of BC-containing aerosols and the absorption enhancement ($E_{abs}$) due to the lensing effect. It is crucial to reasonably represent the mixing of BC with other aerosol components to reduce the uncertainty. In this study, the absorption properties of $PM_{2.5}$ were investigated based on the nested air quality prediction model system (NAQPMS) with different assumptions of the aerosol mixing state. The absorption coefficient ($b_{abs}$) is highest under uniform internal mixing, lower under core-shell mixing, and lowest under the assumption of external mixing. The result under core-shell mixing is closest to the observation. The aging process and coating thickness were well produced by the advanced particle microphysical module (APM) in NAQPMS. Then the fraction of embedded BC and secondary components coating aerosols was used to constrain the mixing state. The $E_{abs}$ at 880 nm over the Beijing-Tianjin-Hebei region was 2.0~2.5 under core-shell mixing. When the fraction of coated BC and the coating layer are resolved, the $E_{abs\_880}$ caused by the lensing effect can decrease by 30~43% to 1.2~1.7, which is close to the range reported in previous studies. This study highlights the importance of representing the microphysical processes governing the mixing state and aging of BC and provides a reference for quantifying its radiative effect.

## 1 Introduction

Aerosols have important environmental and climatic effects, affecting not only transportation and public health but also the global radiation balance (Xu et al., 2013; IPCC, 2021). Mainly originating from incomplete burning, black carbon (BC) is an important component of aerosols. The light absorption of BC particles is enhanced by the "lensing effect" of coating, which affects the heating of the atmosphere by BC (Fuller et al., 1999). When BC is coated by hydrophilic components, it can act





as cloud condensation nuclei affecting cloud and rainfall. Previous studies have demonstrated the important radiative forcing effect of BC (Jacobson, 2013; Bond et al., 2013). However, uncertainty in calculating the optical parameters of BC-containing aerosols still exists. It is challenging to quantify the radiative effect of BC.

The mixing state describes the distribution of properties across a population of particles (Riemer et al., 2019). Externally mixed means that each particle in a population is composed of a single species. Internally mixed means that each particle in the population consists of the same mixture of all chemical species (Stevens and Dastoor, 2019). The aerosol mixing state is dynamic and changes due to several processes, such as emission, new particle formation, transport, condensation, and coagulation processes. Purely internal and external mixing states are rare in the real atmosphere (Bondy et al., 2018). In addition, observations have shown that not all BC particles are coated, and not all secondary aerosols are coated on BC cores (Li et al., 2016). Transmission electron microscope has shown that only a proportion of BC aerosols is embedded (Wang et al., 2021b). The fraction of thickly coated BC in Beijing winter reduced from 48% to 29% from 2012 to 2018 (Wu et al., 2021). The presentation of aerosol mixing states in atmospheric models at different scales was highlighted by Riemer et al. (2019). In many models, aerosols are presented in a few modes with assumed size distribution and the same mixing state and compositions in a mode (Liu et al., 2016; Mann et al., 2010). The sectional approach is used in limited models to represent the mixing state (Yu et al., 2012; Matsui et al., 2014; Matsui, 2016; Yu et al., 2015). Furthermore, particle-resolved models can accurately simulate BC mixing states, but it is applied only to the box and single-column models due to the high computational cost (Riemer et al., 2009; Zaveri et al., 2010; Curtis et al., 2017). Yao et al. (2022) verified the important yet complicated role of mixing state in calculating aerosol optical properties using an ensemble of 1800 aerosol populations from particle-resolved simulations. Trade off the detail presentation and computational cost, most chemical transport models (CTMs) typically simplify aerosol representation by tracking separate aerosol populations rather than individual particle components (Riemer et al., 2019). Mie theory based on a simple fixed mixing state (external, fully internal, or core-shell mixing) assumption is often used in chemical transport models (Li et al., 2020; Gao et al., 2020).

Comparison between different mixing states conducted in previous studies showed that the BC absorbing properties are sensitive to the mixing state assumptions. Curci et al. (2019) found that aerosol optical depth (AOD) is mainly determined by the aerosol mass and only secondarily affected by the mixing state, however, the absorption enhancement ($E_{abs}$) values depended on the mixing assumption made in the model. The underestimation in modeled absorption AOD decreased from 66% in the external mixing case to 43% in the core-shell mixing case (Tuccella et al. 2020). Partial internal mixing is the most likely mixing state of aerosols. The fraction of internally mixed particles can be calculated using the parameterization of Cheng et al. (2012) and Curci et al. (2019). The fraction of core-shell can be parametrized as a function of the bulk volume ratio (Hu et al., 2022). The mixing state index and mass ratio of coating to BC were used to improve the BC mixing state presentation and aerosol particles in the accumulation mode were partitioned into BC-free and BC-containing particles (Shen et al., 2024). $E_{abs}$ calculated in a partial internal mixing state were approximately 10% lower than those from core-shell mixing simulations (Tuccella et al., 2020). However, quantitative investigations considering the evolution of mixing states



based on microphysical properties are limited (Li et al., 2024). Therefore, the fraction of coated BC and the coating fraction of other components based on microphysical processes should be considered in the optical calculation.

Coating thickness and the heterogeneity of the mixing state are proposed to be the main reasons explaining the gap between field observation, lab investigation, and model simulation in light absorption enhancement (Zhao et al., 2021; Fierce et al., 2020; Wang et al., 2021a). The coating fraction significantly influences the absorption of BC particles. The coating thickness of BC particles can be detected by a single-particle soot photometer (SP2), although the lower measurement limits of SP2 result in an overestimation of the concentrations of pure BC (Zhao et al. 2020), the mass ratio of coating to BC core (MR) can highly impact absorption enhancement (Liu et al. 2017; Zhao et al., 2021). The black carbon aging process is an important source of uncertainty in the assessment of its contribution to global warming (Wang et al., 2022). Black carbon can be coated by other aerosols, which increases the complexity of optical properties. Kang et al. (2023) showed that the aging of BC at night in the residual layer can be higher than in the daytime and enhance its light absorption. The aging degree and mixing state can change very quickly in a polluted environment (Peng et al., 2016). An inadequate understanding of the mixing state of BC greatly hinders the assessment of its climate effects (Huang et al., 2023). However, due to this complexity, few three-dimensional (3-D) models sufficiently resolve BC aging processes (Xie et al., 2023; Zhang et al., 2018). A 3-D modeling study found that the aging time of BC varied greatly and showed significant spatial heterogeneity over the polluted areas in China (Chen et al., 2017b).

In this study, the nested air quality prediction model system (NAQPMS) coupled with an advanced particle microphysics module was used to investigate the absorption properties of aerosols over Northern China in November 2018. The simulation of aerosol components was conducted using the 3-D air quality model. Firstly, three ideal mixing states (external, internal mixing, and core-shell mixing) are considered to study the effect of the mixing state by a Flexible Aerosol Optical Depth module (FlexAOD) based on observation and simulation of $PM_{2.5}$ components. Then, the fraction of coated BC based on the aging degree of BC, the fraction of coating, and the detailed microphysics process were considered during the absorption calculation. Finally, the impact of aging degree on light absorption enhancement was investigated.

## 2 Data and methods

### 2.1 Observation data

The study period is from November 1, 2018, to November 30, 2018. Observations of $PM_{2.5}$ components were obtained from China National Environmental Monitoring Centre. The observation site in Beijing (BJ), located at the China National Environmental Monitoring Centre (40º2′N, 116º41′E), is a typical urban site. Water soluble species ($Na^+$, $K^+$, sulfate, nitrate, ammonium, and $Cl^-$) in $PM_{2.5}$ were measured by Gas and Aerosol Collector (GAC), and particles were collected by wet denuder and detected by Ion chromatography. OC and EC are detected by the Thermo optical transmittance method. OC could further be classified into primary organic carbon (POC) and secondary organic carbon (SOC) using a revised





elemental carbon (EC) tracer method (Zhao et al., 2013). Then secondary organic aerosols (SOA) and primary organic aerosols (POA) were calculated with the following equations (Castro et al., 1999):

$$SOC = OC - EC \times (OC/EC)_{min} \tag{1}$$

$$SOA = 1.6 \times SOC \tag{2}$$

$$POA = 1.6 \times OC - SOA \tag{3}$$

In this study, $(OC/EC)_{min}$ calculated from the observed data was 1.16. Organic matter (OM) was assumed to be 1.6 times of OC.

The light extinction parameters were measured at the tower site of the Institute of Atmospheric Physics (IAP), Chinese Academy of Sciences (39°58′28″N, 116°22′16″E) in Beijing. The absorption coefficient at a wavelength of 880nm was directly measured by a seven-wavelength Aethalometer (AE33, Magee Scientific Corp.) (Sun et al., 2021). The extinction coefficient ($b_{ext}$, λ=630 nm) of PM$_{2.5}$ was measured by a cavity-attenuated phase shift extinction monitor (CAPS PMext Aerodyne Research Inc.). The absorption coefficient at 630 nm is derived using a fitted power law relationship at seven wavelengths (Ran et al., 2016). The absorption coefficient at 880 nm was used to analyze the performance of the model as BC is the major contributor to aerosol absorption at 880nm. PM$_{2.5}$, component, and absorption data were used to evaluate the model performance.

### 2.2 Air quality model

NAQPMS is a 3-D Eulerian terrain-following chemical transport model developed by the Institute of Atmospheric Physics, Chinese Academy of Sciences (Wang et al., 2001). The NAQPMS model coupled with an advanced particle microphysics module (APM, Yu and Luo, 2009) was used in this study (Chen et al., 2014). NAQPMS includes physical processes such as advection, convection, diffusion, and deposition, and chemistry processes such as gas-phase chemistry, aqueous chemistry, and aerosol processes. A volatility basis set (VBS) framework for secondary organic aerosols has been coupled to NAQPMS to improve the performance of SOA (Yang et al., 2019; Chen et al., 2021). The APM module includes microphysical processes like nucleation, condensation, evaporation, and coagulation (Yu and Luo, 2009; Chen et al., 2021). Particles are represented by the sectional bin scheme in the APM. Secondary inorganic particles are distributed by 40 bins covering 0.0012–12 μm. BC and OC are represented using 28 bins, and other primary particles such as dust and sea salt are represented by 4 bins. The evolution of particle size distributions and aging processes of BC due to condensation and coagulation are well reproduced by the model (Chen et al., 2017a; Du et al., 2019).

A two-nested model domain was set up in this study. The parent domain covers Northern China at a resolution of 27 km and the second domain covers Beijing-Tianjin-Hebei and surrounding regions with a resolution of 9 km, see Figure 1. To represent fine vertical structures, 30 vertical levels were adopted, including 17 levels below 2 km. The mesoscale model WRF was used to provide the meteorological field for NAQPMS. The initial and boundary conditions for meteorology were provided by NCEP final reanalysis data (FNL) every six hours (https://rda.ucar.edu/datasets/ds083.2/), and MOZART model provided initial chemical fields. The Multi-resolution Emission Inventory for China (MEIC) developed by Tsinghua



University with a resolution of 0.1×0.1° was used (http://meicmodel.org.cn). The base year of the emission inventory was
2018. MEIC covers 10 species including BC, OC, PM$_{2.5}$, PM$_{10}$, CO, NH$_3$, SO$_2$, NOx, CH$_4$, and VOCs. The configuration of
WRF and NAQPMS can be seen in Table S1. The simulation of WRF and NAQPMS started on October 24, 2018, and the
first 7 days were set aside as spin-up time.

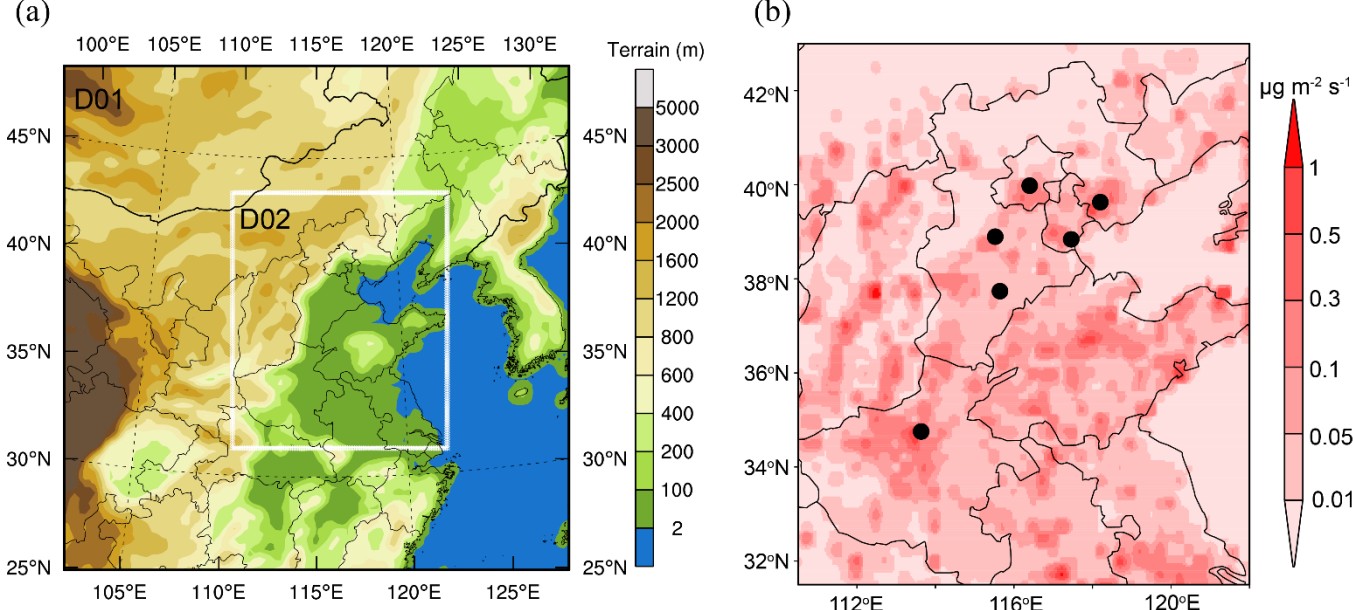

**Figure 1. (a) The model domain. (b) Emission rate of primary PM$_{2.5}$ and the location of observation sites. Black dots are pollution sites.**

### 2.2.1 Flexible Aerosol Optical Depth (FlexAOD) module

In this study, the particles are assumed to be spherical and Mie theory (Mie, 1908) is applied to study their optical properties.
The FlexAOD (http://pumpkin.aquila.infn.it/flexaod/) (Curci et al., 2015) was employed to calculate extinction and single
scatter albedo. The components of PM$_{2.5}$ and relative humidity (RH) at ground level are used as input parameters. The
average volume of particles is computed for each species by dividing mass by the species' density. Mixing states considered
in the study include the external mixing assumption (EXT), internal homogeneous assumption (HOM), and the core-shell
assumption (CS). For the external mixing assumption, extinction is the sum of each species under specific relative humidity.
For internal mixing cases (HOM and CS), the particles conform to a lognormal size distribution. For the internal
homogeneous assumption, the volume average refractive index is a function of particle size over all species. For the core-
shell assumption, the refractive index for BC core and homogeneously mixed shell (secondary inorganic aerosols and
secondary organic aerosols) are calculated separately. The Mie code based on Toon and Ackerman (1981) is used for the
core-shell internal mixing, and the code based on Mishchenko et al. (1999) is adopted for external and homogeneous internal
mixing. The size distribution of the different aerosols is taken from the OPAC (Optical Properties of Aerosols and Clouds)
database (Hess et al., 1998). The mean diameter of BC is assumed to be 30 nm based on Dentener et al. (2006). The density,





complex refractive index, particle hygroscopic growth factor, mean radius, and standard variation of log-normal size distribution are shown in Table S2.

### 2.2.2 Optical module based on APM

APM is a size-resolved, mixing-state-resolved advanced particle microphysics model coupled in NAQPMS. The mixing
state in APM is assumed to be semi-external mixing, which includes internal mixing, external mixing, and core-shell mixing. The seeding particles generated by emission and nucleation (including BC, OC, sulfate, dust, and sea salt) can be coated by secondary particles (including sulfate, nitrate, ammonium, and SOA) through condensation, coagulation, chemical reactions, equilibrium uptake, and hygroscopic growth processes. Sulfate coated by SIA or SOA is considered to be internal mixing. BC, OC, dust, and sea salt coated with SIA or SOA are considered to be core-shell mixing. These coated particles are
externally mixed. The mixing of BC particles with other aerosol components can be well resolved hourly. More details can be seen in Yu and Luo (2009) and Chen et al. (2017b).

When calculating the optical parameters of aerosols, the scheme by Yu et al. (2012) was used. The particles are assumed to be spherical and key particle optical properties including extinction efficiency, single scattering albedo, and asymmetry parameter at each wavelength are calculated by Mie theory based on the core diameter, shell diameter, real and
imaginary components of the refractive index of core and shell. The core-shell code based on Toon and Ackerman (1981) is used in APM. To reduce computation cost, three lookup tables are used: one for particles without solid absorbing cores, the second for coated BC, and the third for coated dust. The volume-averaged refractive indices of species other than BC and dust are calculated based on the composition simulated by NAQPMS. Details can be seen in Yu et al. (2012) and references therein.

### 2.3 Sensitivity test design

In this study, the 3-D chemical transport model NAQPMS coupled with the advanced particle module (APM) was used to reproduce the evolution and spatial distribution of pollutants. The mass concentration, size distribution and mixing state of aerosols are calculated by NAQPMS+APM. FlexAOD is a module that calculates the extinction property of aerosols under different mixing state assumptions based on Mie theory and a fixed size distribution, using the input of aerosol components'
mass concentration and relative humidity as shown in **Sect. 2.2.1**. There are two approaches to calculating optical properties. The absorption property of aerosols can be investigated by FlexAOD with the input of component concentration simulated by NAQPMS+APM and assumed size distribution. The fraction of embedded BC and the traction of coating aerosols calculated by NAQPMS+APM can be used to constrain the mixing state in FlexAOD. In the other approach, the absorption property can be investigated by the optical module based on APM with core and shell information calculated by
NAQPMS+APM as shown in **Sect. 2.2.2**. Then a series of sensitivity tests were designed to explore the impact of mixing state, components, aging process, and detailed microphysical processes (Table 1).





Firstly, to see the effect of mass concentration and mixing state on the optical properties, sensitivity tests with different mixing states (external, internally homogeneous, and core-shell) were conducted using FlexAOD. EXTo, HOMo, and CSo refer to cases calculated using FlexAOD with observed components as input under external, homogeneous internal, and core-shell mixing states, respectively. EXTs, HOMs, and CSs refer to cases calculated using FlexAOD with components simulated by NAQPMS+APM as input under external, homogeneous internal, and core-shell mixing states, respectively. Comparing EXTo, HOMo, and CSo can show the impact of the mixing state. Comparing EXTs, HOMs, and CSs can also show the impact of the mixing state. Comparing $CS_O$ with $CS_S$, the impact of mass concentration on optical properties can be obtained. Secondly, to see the impact of the aging process (fraction of embedded BC core and fraction of coating aerosols), simulations using partial core-shell mixing state in FlexAOD, CS-$F_{in}$ and CS-$F_{in}F_c$, were designed. Additionally, components, size distribution, and mixing state simulated by NAQPMS+APM were used to calculate the optical properties (CS-APM). The impact of the microphysical process can be investigated by comparing CS-$F_{in}F_c$ with CS-APM.

**Table 1 Simulation test design**

| Case | Method | Input | Size distribution | Mixing state |
|---|---|---|---|---|
| EXT$_O$ | FlexAOD | observed | fixed | external |
| HOM$_O$ | FlexAOD | observed | fixed | internal homogeneous |
| CS$_O$ | FlexAOD | observed | fixed | core-shell |
| EXT$_S$ | FlexAOD | simulated | fixed | external |
| HOM$_S$ | FlexAOD | simulated | fixed | internal homogeneous |
| CS$_S$ | FlexAOD | simulated | fixed | core-shell |
| CS-$F_{in}$ | FlexAOD | simulated | fixed | partial core-shell and partial bare BC |
| CS-$F_{in}F_c$ | FlexAOD | simulated | fixed | partial core-shell, partial bare BC and partial coating aerosols |
| CS-APM | APM | simulated | simulated | semi-external (hourly) |

| Impact | Description |
|---|---|
| EXT$_O$ vs. HOMo vs. CSo | Impact of mixing state when inputting observed data |
| EXT$_S$ vs. HOM$_S$ vs. CS$_S$ | Impact of mixing state when inputting simulated data |
| CSo vs. CSs | Impact of aerosol mass concentration |
| CSs vs. CS-$F_{in}$ | Impact of aging process (fraction of embedded BC) |





| CSs vs. CS-$F_{in}F_c$ | Impact of the aging process (fraction of embedded BC and coating shell) |
|---|---|
| CS-$F_{in}F_c$ vs. CS-APM | Impact of detailed microphysical process |

## 2.4 Model evaluation

Statistical parameters such as the correlation coefficient (R), normalized mean bias (NMB), index of agreement (IOA), the fraction of the simulations within a factor of two of the observations (FAC2), mean fractional bias (MFB) and mean fractional error (MFE) were used in this study to evaluate the performance of NAQPMS (Table S3). NAQPMS reproduced the temporal distribution of $PM_{2.5}$ in Beijing well (Fig. 2). As shown in Table S4, the R between the observed and simulated hourly $PM_{2.5}$ concentrations of six sites over Beijing-Tianjin-Hebei and surrounding regions were within 0.55~0.76. The

NMB was within -0.22~0.13, which satisfied the model performance criteria proposed by Emery et al. (2016). There was only a small overestimation of 13% in the simulated $PM_{2.5}$ in Zhengzhou. And the IOA reached more than 0.72. The MFB and MFE of $PM_{2.5}$ in all six sites were within the benchmarks, which satisfied the model performance criteria proposed by Boylan and Russell (2006).



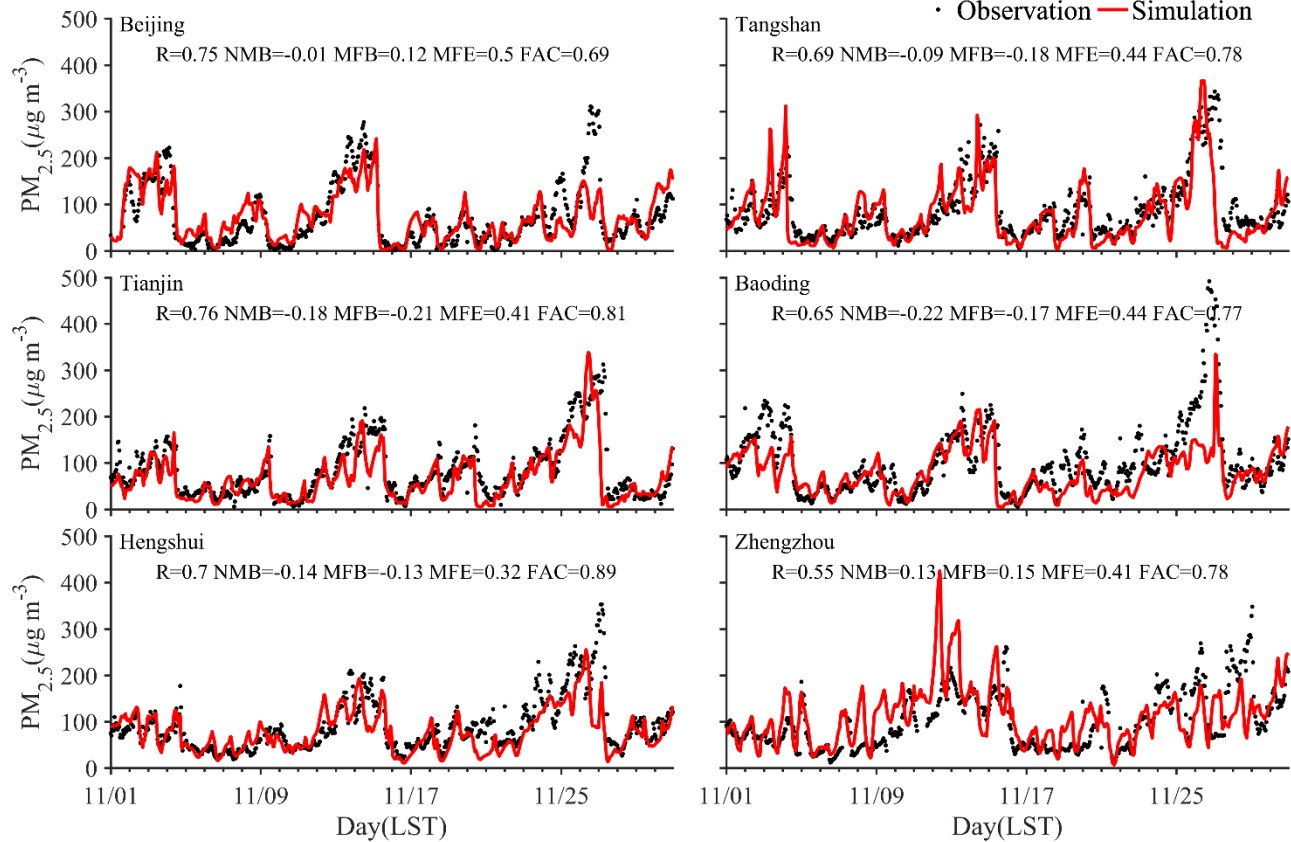

**Figure 2. Model evaluation of PM$_{2.5}$ at eight sites in the Beijing-Tianjin-Hebei region in November 2018.**

NAQPMS also exhibited good performance in representing the PM$_{2.5}$ components in Beijing (Fig. 3). The R values between the observed and simulated nitrate, sulfate, ammonium, element carbon, primary organic carbon, and secondary organic carbon were 0.74, 0.83, 0.74, 0.55, 0.49, and 0.47 in Beijing, respectively. However, the simulation of secondary inorganic aerosols was underestimated by -62%~-8%. This is likely caused by insufficient heterogeneous formation of sulfate and nitrate (Li et al., 2018). Black carbon and primary organic aerosols were overestimated by 7.2% and 26.1%, which is probably related to the emission inventory. Monthly mean emissions were used in this study and there is substantial uncertainty in emission inventory (Li et al., 2017). However, the reasonable simulation of aerosol mass concentration lays a solid foundation for simulating the optical properties of BC-containing aerosols.



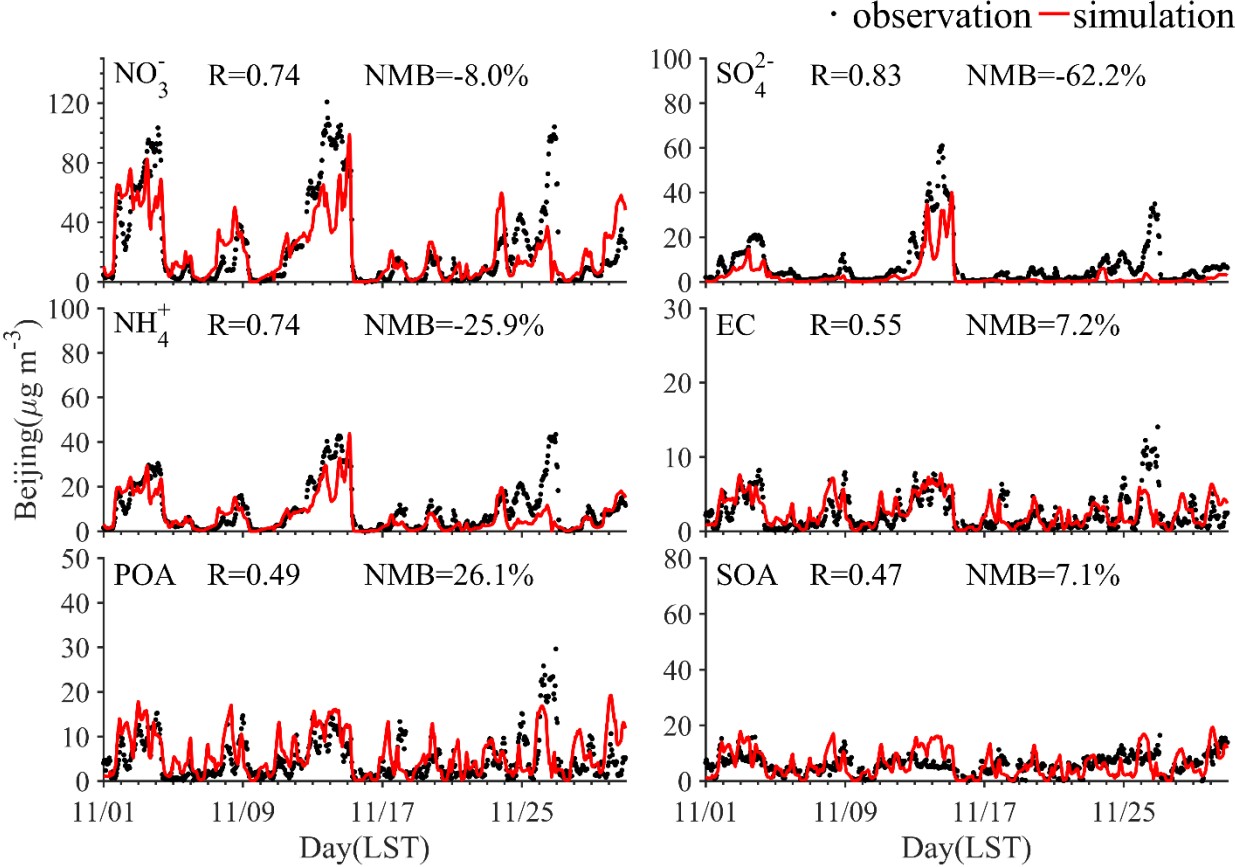

**Figure 3. Simulated and observed PM$_{2.5}$ components in Beijing**

## 3 Results

### 3.1 Absorption properties based on observed components

#### 3.1.1 Description of observations

The time series and proportion of various chemical components of PM$_{2.5}$ are shown in Fig. 4. During the study period, the average mass concentration of PM$_{2.5}$ was 74.4 ± 68.7 μg m$^{-3}$. Nitrate is the main component of PM$_{2.5}$, accounting for 36.4% on average, followed by organic matter, ammonium, and sulfate, accounting for 16.6%, 15.4% and 11.5%, respectively. EC, crustal elements, and chloride salt accounted for 3.9%, 8.8%, and 4.1%, respectively. The average RH during the period was 39 ± 17.9% and the temperature was 8.3 ± 3.2 °C. The average $b_{sca}$ (±1σ) and $b_{abs}$ (±1σ) at 630 nm during the study period were 169.1 ± 212.3 Mm$^{-1}$ and 46.5 ± 48.5 Mm$^{-1}$ in Beijing, respectively. The average $b_{abs}$ (±1σ) at 880 nm during the study period were 30.7 ± 25.2 Mm$^{-1}$. The decrease in visibility is mainly caused by particle scattering extinction. The $b_{sca}$ and $b_{abs}$ in this study were much lower than those observed in Beijing in the winter of 2016 (Xie et al., 2019), but the $b_{ext}$ in this study





was higher than that observed in Beijing in the winter of 2019 (Sun et al., 2021), indicating that the decreases in $PM_{2.5}$ in recent years also caused similar reductions in extinction coefficients.

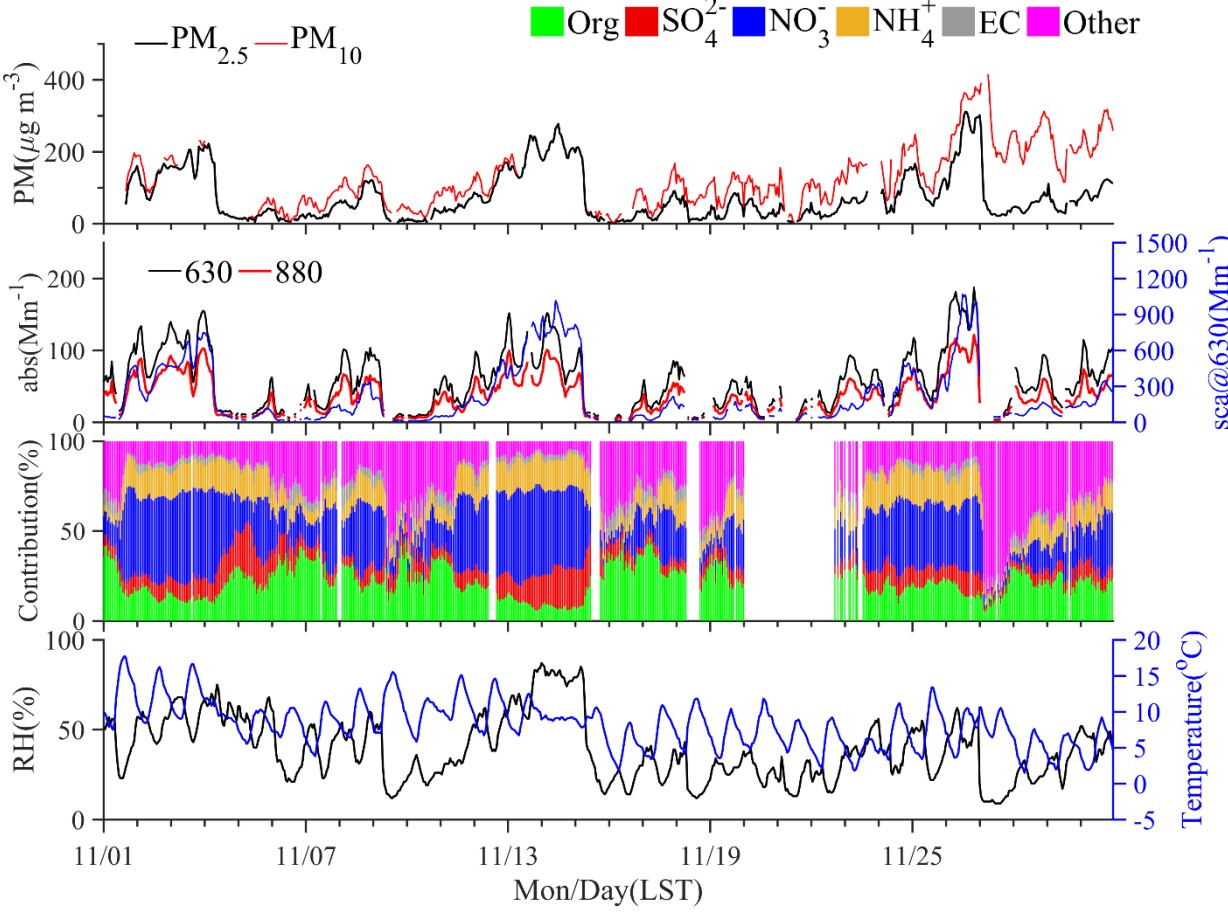

**Figure 4. Evolution of observed PM₂.₅, components, aerosol extinction coefficient, and meteorology parameters in November 2018 at Beijing**

### 3.1.2 Absorption coefficient calculated by FlexAOD based on observation

The observed $PM_{2.5}$ components and RH were used to calculate the optical properties using FexAOD. It should be noted that the times missing component data were excluded when calculating optical properties. Comparison between the observed and simulated absorption coefficient showed that the simulations by FlexAOD under the three mixing state assumptions are highly correlated with the observation, and the correlated coefficient can reach 0.88. However, using different mixing state assumptions led to widely varying results, see Figure 5. On average, the $b_{abs}$ of 880 nm calculated for the core-shell mixing state was 2.3 times higher than that of external mixing. For the external mixing state, the calculation was underestimated by 59%, while it was overestimated by 79% under uniform internal mixing. The simulation for core-shell mixing was closest to observations, with an underestimation of 4.7%. The absorption coefficient under uniform internal mixing is the highest,




followed by core-shell mixing and the calculation for external mixing is the lowest. This is consistent with the findings of Curici et al. (2019). The assumption of internal mixing or external mixing is not realistic. Partial internal mixing with partial coating is closer to reality and it should be considered for absorption calculations as reported by Curci et al. (2019).

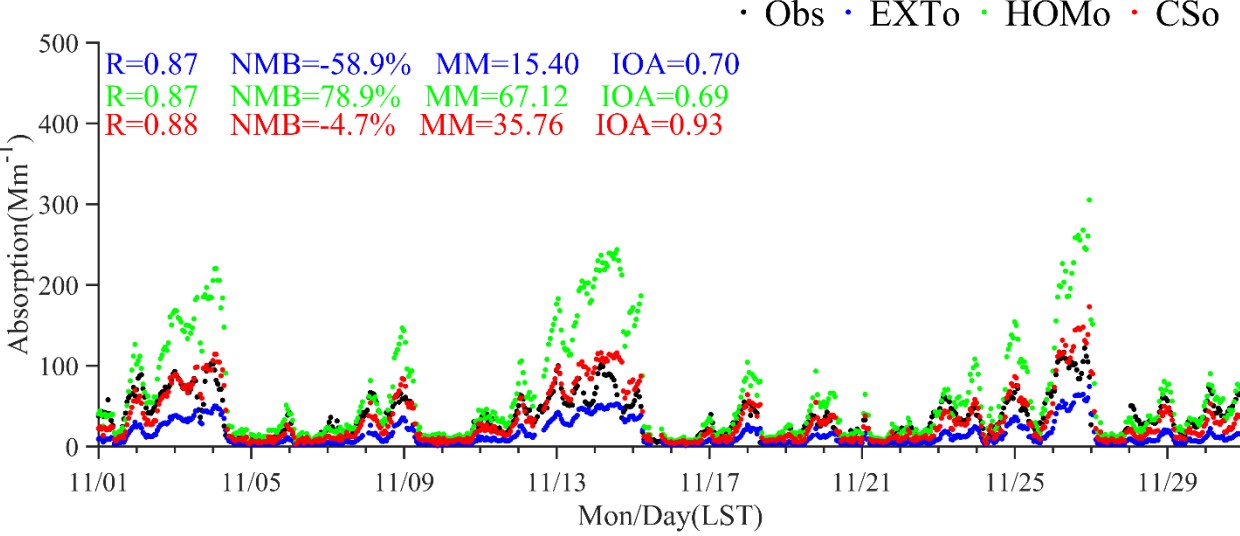

**Figure 5. Observed absorption and calculated absorption at 880nm under external mixing (EXTo), homogeneous internal (HOMo), and core-shell mixing (CSo) by FlexAOD based on observed components.**

### 3.2 Absorption property based on simulations by NAQPMS

Comparison of the absorption coefficients between observation and calculation with FlexAOD based on PM$_{2.5}$ simulation from NAQPMS with different mixing states can be seen in Table 2. In this study, only times in which the simulated PM$_{2.5}$ was within a factor of two of observation were considered in the optical calculation. The average of observed $b_{abs}$ is 43 Mm$^{-1}$. There were large variations in the absorption coefficient under different mixing states. In the EXTs case, the absorption coefficient at 880m was underestimated by 54%, and only 40% of the modeled values were within a factor of 2 of the observations. In the HOMs case, the absorption at 880m was overestimated by 95%, and 58% of the modeled values were within a factor of two of observations. In the CS$_S$ case, FAC2 increased to 0.93 compared with EXTs simulations, and the model overestimated the absorption coefficient by 10%. This was related to the overestimation of black carbon. In this study, black carbon and POA are assumed to have light absorption properties and only BC particles act as cores in FlexAOD. The core-shell mixing state produces the most accurate results compared to observations. Comparing the results of CS$_S$ with CS$_O$ (where calculations with FlexAOD were based on observed components under core-shell mixing state), the results show that the effect of simulated components on the absorption coefficient can be up to 15%, which is much smaller than the impact of the mixing state.





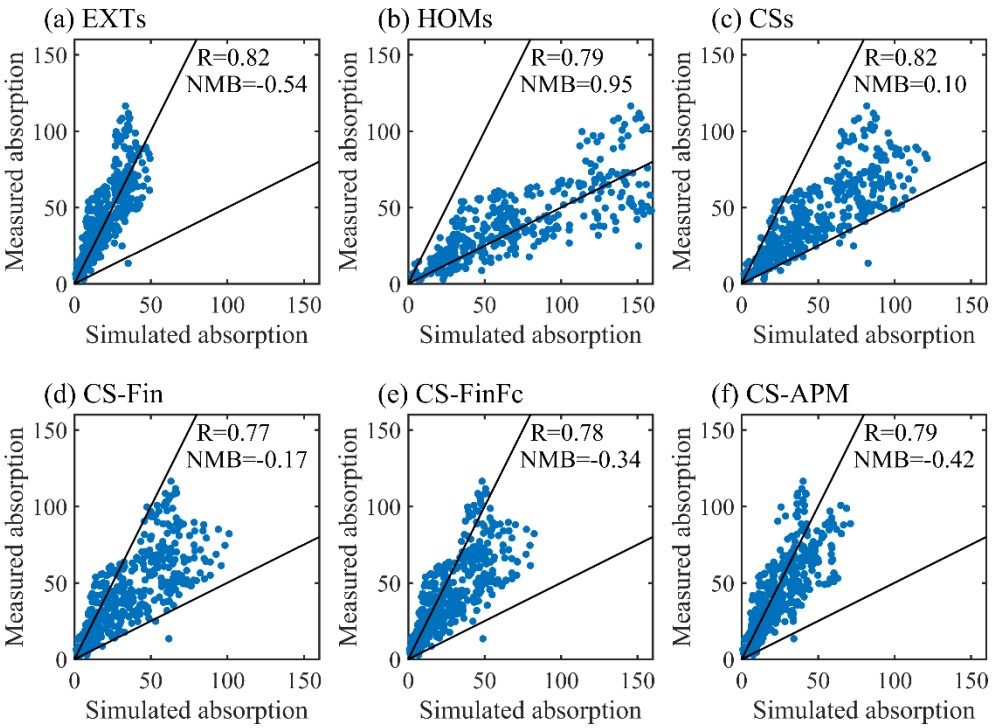

**Figure 6. Comparison of observed and simulated absorption coefficient at 880 nm under different mixing states (EXT$_S$, HOM$_S$, CS$_S$, CS-F$_{in}$, CS-F$_{in}$F$_c$, and CS-APM) at IAP in Beijing.**

**Table 2. Intercomparison of the performance of absorption coefficient at 880 nm under different mixing states. b is the ratio of**
265 **simulation to observation.**

| Schemes | b | R | NMB | FAC2 |
|---|---|---|---|---|
| EXTs | 0.46 | 0.82 | -0.54 | 0.40 |
| HOMs | 1.95 | 0.79 | 0.95 | 0.58 |
| CSs | 1.10 | 0.82 | 0.10 | 0.93 |
| CS-F$_{in}$ | 0.83 | 0.77 | -0.17 | 0.84 |
| CS-F$_{in}$F$_c$ | 0.66 | 0.78 | -0.34 | 0.66 |
| CS-APM | 0.58 | 0.79 | -0.42 | 0.72 |



**3.3 Constraint of the fraction of embedded BC and secondary components coating aerosols**

In the real world, the mixing state of particles is complex. Wang et al. (2021b) using an electron microscope found that the embedded fraction of BC significantly influenced the absorption. In the extremely polluted winter period of January 2013, more than half of BC particles were thickly coated by non-refractory materials (Wu et al., 2016). Along with the implementation of the Air Pollution Prevention and Control Action Plan, the mass of BC and the fraction of thickly coated BC changed (Wu et al., 2021). Cheng et al. (2012) proposed that the fraction of internally mixed particles can be parameterized based on oxidized nitrogen oxides and total reactive nitrogen. Curci et al. (2019) used the mass ratio of secondary inorganic aerosols and organics to BC as the fraction of internally mixed particles.

Emitted hydrophobic black carbon becomes hydrophilic due to aging processes. In this study, the aging of BC can be resolved by NAQPMS+APM. The detailed aging processes of aerosols are considered in a physical manner. The model represents the aging processes by simulating condensation and coagulation. The ratio of hydrophilic BC to total BC is used as a proxy for the fraction of embedded BC. The evolution of the ratio of particle diameter to BC-core size (Dp/Dc), the fraction of embedded BC ($F_{in}$), and the fraction of secondary components coating on BC ($F_c$) at Beijing is shown in Fig. 7. When $F_{in}$ is equal to 0, it means the BC is externally mixed with other aerosols, and when $F_{in}$ is equal to 1, it means all BC particles are coated by other aerosols. The average ratio of $F_{in}$ during the study period in Beijing is 34.1%, which is a bit lower than the ratio of 0.48 and 0.63 obtained by the method of Cheng et al. (2012) and Curci et al. (2019), respectively. Also, $F_{in}$ in this study is closely related to the ratio of secondary inorganic aerosols to BC, with an R of 0.72. Zheng et al. (2022) also found that secondary inorganic aerosols dominated the light absorption enhancement with online observational datasets. We consider a separate case, CS-$F_{in}$, where the $F_{in}$ fraction of BC particles is core-shell mixed with other aerosols, and a 1-$F_{in}$ fraction of BC particles is bare and external mixing. The calculated BC absorption at 880 nm ($b_{abs\_880}$) for the CS-$F_{in}$ case was 35.5 Mm$^{-1}$, which was close to the measured mean value, and 84% of simulations of absorption coefficients were within a factor of two of the observations.





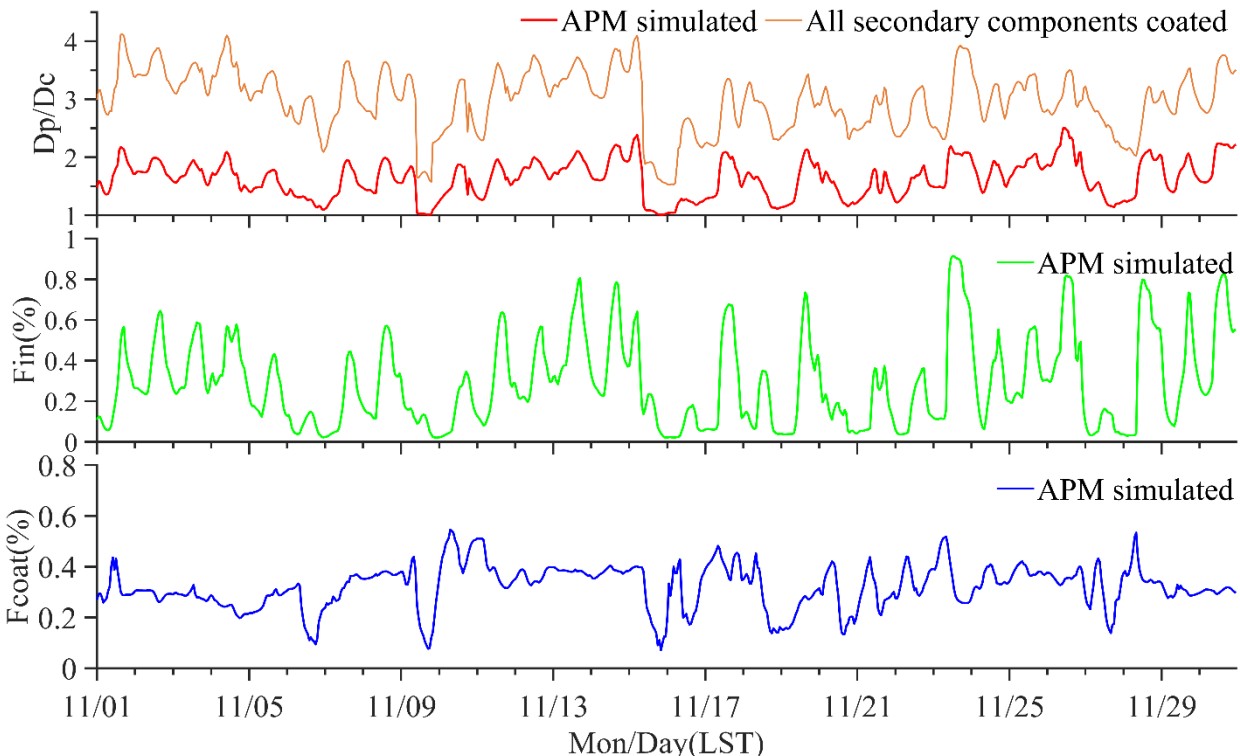

**Figure 7. Evolution of ratio of particle diameter to BC-core size (Dp/Dc), the fraction of embedded BC (Fin), and the fraction of secondary components coating on BC (Fc) at IAP in Beijing. "APM simulated" refers to parameters simulated by the advanced particle microphysics module in NAQPMS.**

Aerosols in the atmosphere include BC-containing aerosols (coated BC and bare BC) and BC-free aerosols (Zhao et al., 2022). In this study, the fraction of secondary aerosol coating on BC is also considered in the optical calculation. As sulfate aerosols include sulfate coating on BC, OC, dust, and sea salts, then the fraction of sulfate coating on BC was used as the fraction of secondary components coating on BC ($F_c$). When $F_c$ is equal to 0, there is no coating over BC particles, and when $F_c$ is equal to 1, it means all other aerosols are coated on BC. We thus consider a scenario CS-$F_{in}F_c$ where $F_{in}$ fraction of BC particles are core-shell mixed and the $F_c$ fraction of secondary components and other aerosols are externally mixed. The average $F_c$ during the study period in Beijing is 34.3%. The calculated BC absorption at 880 nm ($b_{abs\_880}$) for the CS-$F_{in}F_c$ case was 28.1 Mm$^{-1}$, and 66% simulations of absorption coefficients were within a factor of two of observations.

As described in **section 2.2.2**, optical properties were calculated based on Mie theory using the core and shell information calculated by APM considering microphysical processes. The observed absorption coefficient and that simulated by NAQPMS+APM under a semi-external mixing state, namely CS-APM, is shown in Table 2 and Fig. 6f. The results show that the simulated absorption at 880nm matches the observation reasonably well, with an R of 0.79, although there was an underestimation with NMB of 0.42. The CS-$F_{in}F_c$ case considers the fraction of embedded BC and the fraction of secondary components coating BC calculated with APM. Comparison between CS-$F_{in}F_c$ and CS-APM showed the impact of




considering the detailed microphysics process on absorption property. The FAC2 of 0.72 in the CS-APM case is greater than 0.66 in the CS-$F_{in}F_c$ case. The underestimation of 42% in CS-APM is larger than the 34% in CS-$F_{in}F_c$. This underestimation can be attributed to the assumed morphology of BC-containing particles, the size distribution of primary particles input to APM, and the concentration of secondary components coated on BC. As shown in Fig. 3, there is an underestimation of 8-62% in the simulation of secondary inorganic aerosols. Even if the mode of the physical process is correct, the coating on BC can be underestimated, which affects the absorption characteristics of aerosols.

### 3.4 Light absorption enhancement due to mixing state

The light absorption enhancement is the ratio of the light absorption coefficient of coated BC and bare BC. $E_{abs}$ is proposed to quantify the lensing effects, however, large uncertainty exists in $E_{abs}$ and the radiative effect of black carbon.

### 3.4.1 The impact of the detailed microphysics process on absorption enhancement

We modified the APM module in NAQPMS so that BC does not mix with other chemical species in the calculation of the microphysics process and optical properties. This sensitivity test was conducted by turning off the coating process in APM. The radiative absorption enhancement was the ratio of the absorption coefficient in the base simulation to that in the sensitivity test.

The mass ratio of the coating of BC to BC (MR) can be used to represent the aging degree (Du et al., 2019; Wang et al., 2019). To compare with previous studies, $E_{abs}$ at 630nm is shown in Fig. 8. The evolution of $E_{abs}$ and MR shows that $E_{abs}$ is positively correlated to MR, and the R can reach 0.88. This is consistent with Liu et al. (2017) who showed that $E_{abs}$ is closely related to MR. Under the same MR, $E_{abs}$ can vary by 0.49. When MR equals 3, $E_{abs}$ varied by 0.25. The $E_{abs}$ in the CS-APM case in Beijing is much higher than the measurement in Taizhou (Zhao et al. 2021). However, the measurements in Beijing by Xie et al. (2019) fall in the range of this study when MR is less than 5. $E_{abs}$ in the CS-APM case is higher than that from the laboratory study in Peng et al (2016) when MR is less than 3, but it is lower when MR is bigger than 5.

The spatial distribution of $E_{abs}$ at 880nm is shown in Fig. 9d. The $E_{abs\_880}$ over Beijing-Tianjin-Hebei and the surrounding region was about 1.3~1.8. The $E_{abs\_880}$ in the CS-APM case is a bit higher than that in the CS-$F_{in}F_c$ case. The spatial distribution of $E_{abs\_880}$ also showed lower values of 1.3~1.7 over the source region and higher values of 1.6~1.8 over the outflow region. The average $E_{abs}$ at Beijing at 630nm and 880nm from APM and Mie theory are 1.58 and 1.55.

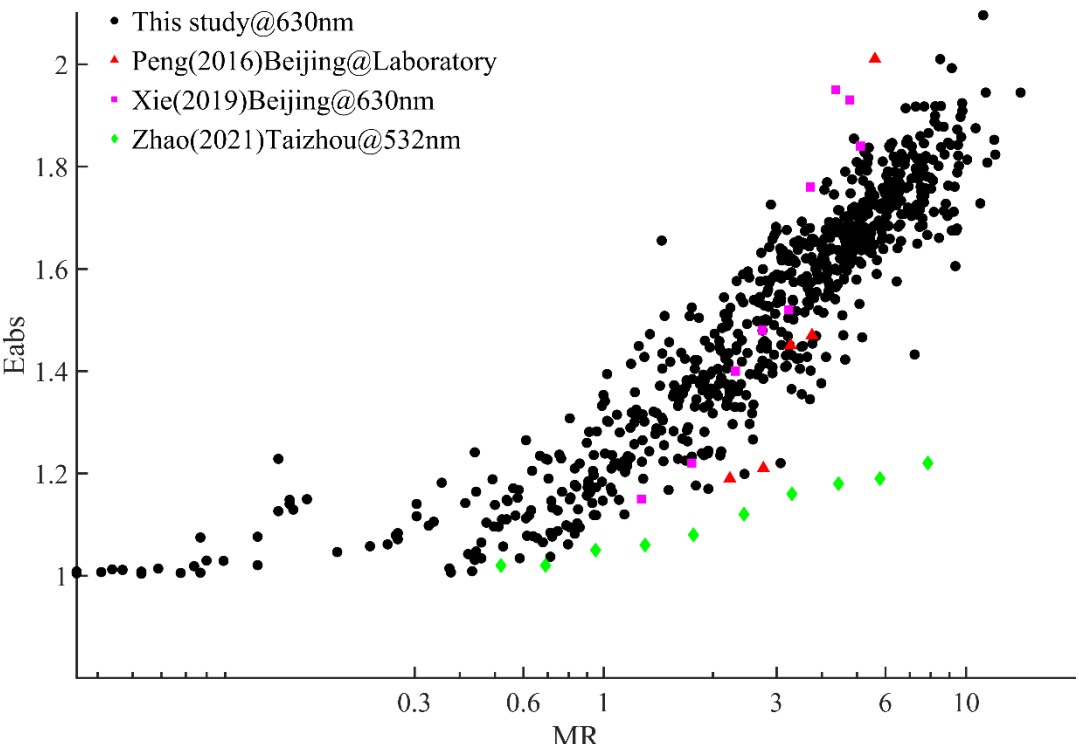

**Figure 8. The absorption enhancement in CS-APM case under different mass ratios (MR) of coating materials and BC core**

**3.4.2 Impact of the aging processes on light absorption enhancement**

The spatial distribution of $E_{abs}$ at 880nm is shown in Fig. 9. The spatial distribution of the absorption enhancement in CS-Fin

and CS-$F_{in}F_c$ cases showed that $E_{abs}$ was lower near the emission source and higher in the outflow region (Bohai and Yellow Sea, Taihang mountain). This is because BC was aged by condensation and coagulation processes during transport in the atmosphere. As shown, the values of $E_{abs\_880}$ over Beijing-Tianjin-Hebei and the surrounding region from FlexAOD under the core-shell mixing state are about 2.0~2.5. After considering the fraction of embedded BC, the $E_{abs}$ decreased to 1.3~2.1, representing a decrease of 11%~34%. Considering the fraction of embedded BC and the fraction of coating, the $E_{abs}$

decreased to 1.2~1.7, representing a decrease of 30%~43%. The values of $E_{abs}$ in the CS-$F_{in}F_c$ case are 1.2~1.5 near the emission sources and 1.5~1.7 over the outflow region. These values are similar to the currently accepted range of 1.2–1.6 (Bond et al., 2013; Matsui et al., 2016; Liu et al., 2017; Curci et al., 2019). The distribution of average $E_{abs}$ and SSA values with height in Beijing is shown in Fig. 10. $E_{abs}$ increased with height while SSA decreased with height in CS-$F_{in}$ and CS-$F_{in}F_c$ cases. Relatively low $E_{abs}$ values (1.3~1.6) are concentrated in layers below 500 m. This is related to the low-level

anthropogenic emission and the ability of BC in the upper layer to be transported over wider regions.





**Figure 9. The absorption enhancement at 880 nm in (a) core-shell mixing (CS), (b) CS-F$_{in}$, (c) CS-F$_{in}$F$_c$, and (d) CS-APM. And changes in radiation absorption enhancement in (e) CS-F$_{in}$ and (f) CS-F$_{in}$F$_c$ cases compared with core-shell mixing.**





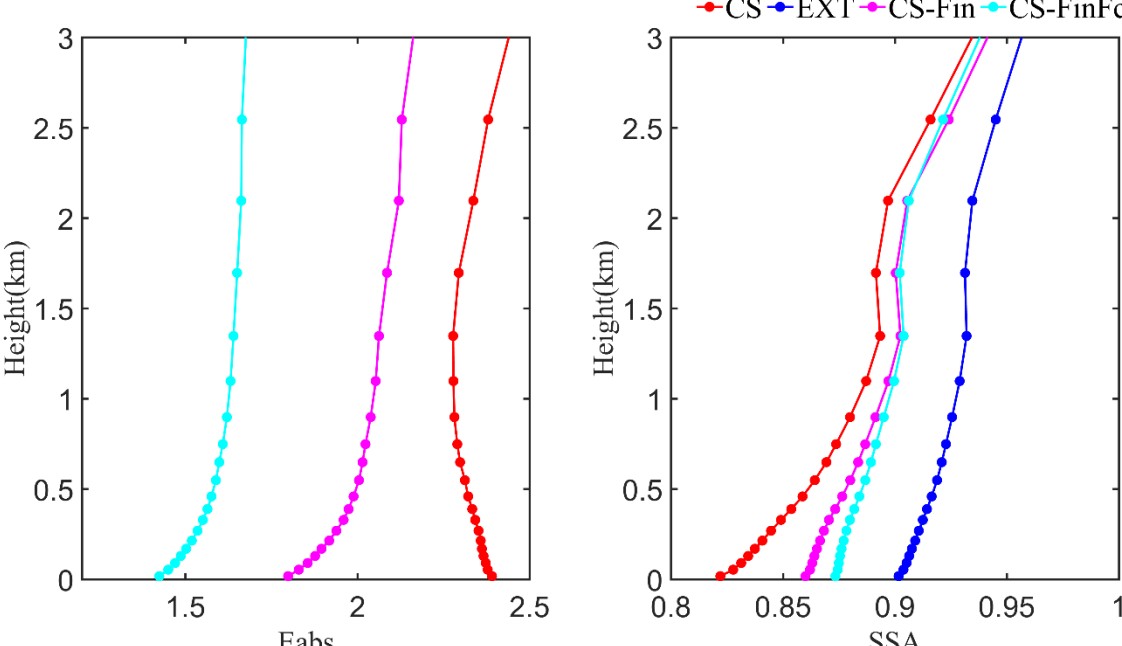

**Figure 10. The distribution of $E_{abs}$ and SSA values with height**

In Beijing, the absorption enhancement at 630 nm and 880 nm is 2.65 and 2.39 for the core-shell mixing state using FlexAOD (Fig. 11). When considering the fraction of embedded BC, the $E_{abs}$ in the CS-$F_{in}$ case at 630 nm and 880 nm are 1.94 and 1.80, decreasing by 26.7% and 24.5% compared to the CS case, respectively. If the fraction of secondary aerosol coating on BC is also considered at the same time, the $E_{abs}$ in the CS-$F_{in}F_c$ case at 630 nm and 880 nm are 1.51 and 1.43, decreasing by 43% and 40.2% compared to the CS case, respectively. Therefore, considering the fraction of secondary aerosol coating on BC, $E_{abs}$ at 630 nm and 880 nm can decrease by 16.2% and 15.7%, respectively, compared to the CS case. The ratios of $E_{abs}$ at 630 and $E_{abs}$ at 880 nm by FlexAOD in this study were both less than 1. This is consistent with the fact that $E_{abs}$ are expected to decrease with increasing wavelength (Liu et al., 2018).





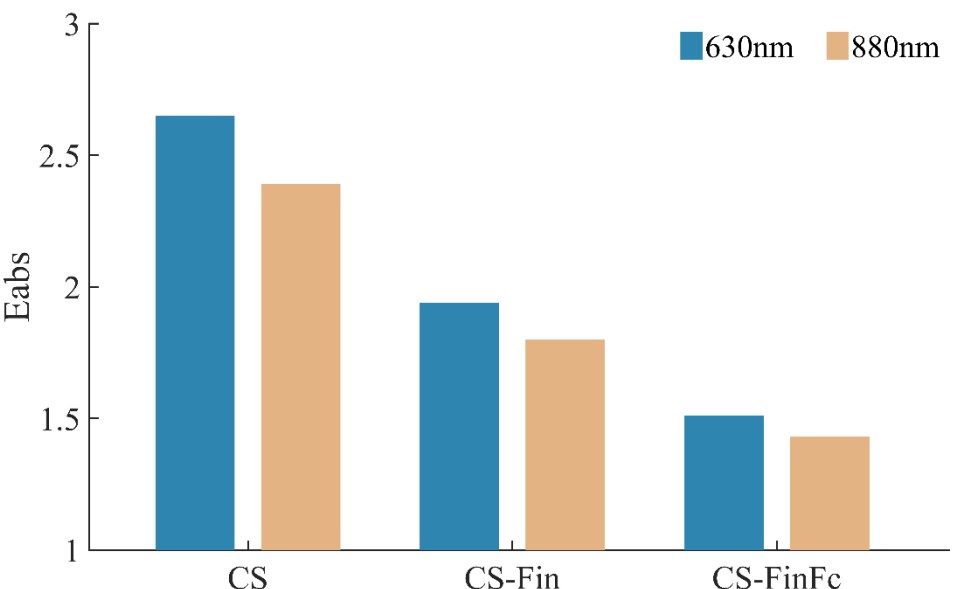

**Figure 11. The radiation absorption enhancement of 630nm and 880nm under different mixing state assumptions in Beijing.**

Comparing the $E_{abs}$ obtained in this study under different mixing states with previous studies, the $E_{abs}$ in CS-$F_{in}$ and CS-$F_{in}F_{c}$ considering the aging process were a bit higher than other laboratory and ambient measurement studies in Beijing (1.03~1.3) (Wang et al., 2019). Sun et al. (2021) using the thermodenuder (TD) method found that $E_{abs}$ at 870nm at an urban site in Beijing was 1.24 ± 0.15. Zhang et al. (2021) using the mass absorption cross section (MAC) method by SP2 found that $E_{abs}$ at 880nm at a rural site in Gucheng was 1.33 ± 0.57. Cui et al. (2016) found that the $E_{abs}$ increased from 1.4 during fresh combustions to approximately 3 for aged BC at a rural site on the North China Plain. But the results for CS-$F_{in}$, CS-$F_{in}F_{c}$, and CS-APM were lower than other model simulations (Curci et al., 2019; Tuccella et al., 2020). The results show that considering the aging process of BC has a significant effect on the absorption enhancement, and should be considered during model $E_{abs}$ calculation.

**4 Summary and conclusions**

Black carbon-containing aerosols have a significant impact on global warming. However, the extent of the impacts is highly uncertain. Component concentration, mixing state, and aging processes are important parameters. In this study, observed and simulated concentrations of $PM_{2.5}$ components in November 2018 are used with Mie theory to investigate the impact of the mixing state and aging process on the light absorption properties of aerosols.

Through a series of sensitivity tests, a systematic comparison was conducted to explore the impact of components, mixing state, aging process, and detailed microphysics on absorption property. Under the same mixing state with observed





and simulated components, $b_{abs}$ can be highly impacted by the simulated concentration of $PM_{2.5}$ components. Sensitivity tests with different mixing states (external, internally homogeneous, and core-shell) using FlexAOD showed that different mixing state assumptions led to widely varying results. The absorption coefficient is the highest under uniform internal mixing, lower under core-shell mixing, which is closest to observation, and lowest under external mixing.

Partial internal mixing and partial coating are the closest to reality. The detailed microphysical processes can be resolved by an advanced particle microphysics module in NAQPMS. The ratio of hydrophilic BC and total BC is used as a proxy for the fraction of embedded BC and the fraction of sulfate coating on BC is used as a proxy for the fraction of secondary components coating on BC. Then the fraction of embedded BC and secondary components coating aerosols was used to constrain the mixing state. Considering the fraction of embedded BC and secondary components coating on BC, the mixing state is closer to reality and the simulation of absorption is also acceptable. The NMB of the simulated absorption coefficient has changed from 10% to -34% in Beijing, and the R changed from 0.82 to 0.78.

Accounting for the aging process of BC has a significant effect on radiative absorption enhancement. The $E_{abs}$ at 880 nm over the Beijing-Tianjin-Hebei area reduced from 2.0~2.5 under core-shell mixing state to 1.3~2.1 when considering the fraction of embedded BC, and to 1.2~1.7, a decrease of 30%~42%, when considering the fraction of embedded BC and the fraction of coating. Considering the detailed microphysical processes, $E_{abs}$ in the CS-APM case was positively correlated with MR with an R of 0.88. The $E_{abs}$ values in CS-$F_{in}F_c$ and CS-APM cases were a bit higher than those from other laboratory and ambient measurement studies in Beijing, but were within the range of previous studies.

The optical property can be affected by uncertainties in the size distribution of primary particle emission (Zhou et al. 2012; Matsui, 2016). Geometric radius and standard variation are two important parameters of size distribution. The optical depth of mineral dust and organic was sensitive to standard variation (Obiso and Jorba, 2018). There is a sector and spatial difference in the size distribution of primary emission (Paasonen et al., 2016). Sensitivity tests should be conducted to see the impact of size distribution on $\sigma_{abs}$ and $E_{abs}$ in future studies. More efforts considering the morphology and the absorption characteristics of coating can also help understand the radiative effect of BC-containing aerosols (Liu et al., 2020; Li et al., 2024).

Overall, this study underscores the importance of the representation of microphysical processes related to BC aerosols and their mixing state. Our results indicate that resolving the fraction of coated BC and the coating layer can significantly impact the calculated $E_{abs}$. Although modeling the mixing state and microphysics process is a challenge for the chemical transport model, the fraction of aged BC and coating aerosols can be used to constrain the mixing state. This study provides a reference for simulating the radiative effect of black carbon aerosols using three-dimensional models.

**Data availability.**

The $PM_{2.5}$ observation data can be obtained from the China National Environmental Monitoring Centre (https://air.cnemc.cn:18007/). The simulated data of this study are available upon request to the corresponding author.



**Author contributions.**

HD, JL and XC designed the work. HD performed the simulation and analysis. Gabriele C and ZFW provided the software. YS, XD and SG processed the measurement data. ZW, WY and LW validated the simulated data. HD wrote the original draft with assistance from co-authors. JL, XC and FY reviewed and edited the manuscript.

**Competing Interests**

At least one of the (co-)authors is a member of the editorial board of Atmospheric Chemistry and Physics.

**Acknowledgements**

We thank the support of the Technological Infrastructure project "Earth System Science Numerical Simulator Facility" (EarthLab). The FlexAOD code can be provided upon request to gabriele.curci@aquila.infn.it.

**Financial support.**

This research was supported by the National Key R&D Program of the Ministry of Science and Technology, China (grant no. 2022YFC3700703, 2020YFA0607803), the Strategic Priority Research Program (B) of the Chinese Academy of Sciences (XDB0760300), the National Natural Science Foundation of China (NSFC) research project (42207133, 42377105).

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
