# Peer review of "Modeling simulation of aerosol light absorption over the Beijing-Tianjin-Hebei region: the impact of mixing state and aging processes"

_EGUsphere, 2024_

## Author Comment (AC1)

Response to comments from reviewers on "Modeling simulation of aerosol light absorption over the Beijing-Tianjin-Hebei region: the impact of mixing state and aging process" by Huiyun Du et al.

We thank the reviewer for the valuable comments and constructive suggestions. We have made extensive corrections to our previous manuscript and responded to the concerns below (in blue).

Reviewer #1

This manuscript investigated the influences of mixing state on BC light absorption properties, mainly based on simulations. Different scenarios were designed, and the modeling results were constrained using field observational data. Although the topic of this manuscript is within the scope of ACP, I have substantial concerns on the methodologies as well as some results. It needs to be re-reviewed after major revisions.

First, the estimation of POC and SOC using the EC-tracer method. In the current manuscript, the minimum OC to EC ratio (1.16) was used to represent primary emissions (Page 4, equation 1). This approach needs to be refined, e.g., by using the lowest 10 % percentile of ambient OC/EC ratios (please refer to https://doi.org/10.1029/2008JD010902; Atmos. Chem. Phys., 15, 2969–2983, 2015; etc.).

**Response:** We acknowledge that this approach may require further refinement. To improve the accuracy of our methodology, we use the lowest 10% percentile of ambient OC/EC ratios, as suggested in the referenced study (Zheng et al., 2015; Lin et al., 2009). We appreciate your guidance on this matter and have revised our methods accordingly.

The figure below shows the estimated intercept and slope, i.e., values of $(OC/EC)_{pri}$, by using the lowest 10% percentile of ambient OC/EC ratios (about 70 data). Corresponding modifications have been made to Figure 3 and Figure 5.

[Figure]

**Figure 1.** Estimation of SOC with EC-tracer method. Blue squares indicate data used to calculate primary OC/EC, while orange-filled circles indicate other OC/EC data.

Zheng, G. J., Duan, F. K., Su, H., Ma, Y. L., Cheng, Y., Zheng, B., Zhang, Q., Huang, T., Kimoto, T., Chang, D., Pöschl, U., Cheng, Y. F., and He, K. B.: Exploring the severe winter haze in Beijing: the impact of synoptic weather, regional transport and heterogeneous reactions, Atmos. Chem. Phys., 15, 2969-2983, 10.5194/acp-15-2969-2015, 2015.

Lin, P., Hu, M., Deng, Z., Slanina, J., Han, S., Kondo, Y., Takegawa, N., Miyazaki, Y., Zhao, Y., and Sugimoto, N.: Seasonal and diurnal variations of organic carbon in PM2.5 in Beijing and the estimation of secondary organic carbon, Journal of Geophysical Research: Atmospheres, 114, https://doi.org/10.1029/2008JD010902, 2009.

**Change in the manuscript:**

OC could further be classified into primary organic carbon (POC) and secondary organic carbon using the elemental carbon (EC) tracer method (Castro et al., 1999, Zhao et al., 2013). A refined EC tracer method was proposed (Zheng et al., 2015; Lin et al., 2009), and data from the lowest 10% percentile of ambient OC/EC ratios was utilized to estimate the primary OC/EC ratio with the following equations (Figure S1).

$$POC = EC \times (OC/EC)_{pri} + N \tag{1}$$

$$SOC = OC - POC \tag{2}$$

[Figure]

**Figure S1. Estimation of SOC with EC-tracer method. Blue squares indicate data used to calculate primary OC/EC, while orange-filled circles indicate other OC/EC data.**

[Figure]

**Figure 1. Simulated and observed PM$_{2.5}$ components in Beijing**

[Figure]

**Figure 2. Observed absorption and calculated absorption at 880nm under external mixing (EXTo), homogeneous internal (HOMo), and core-shell mixing (CSo) by FlexAOD based on observed components.**
Please refer to Lines 100-105, Line 217, 219, Lines 247-251.

In addition, the POC and SOC results should be carefully evaluated, e.g., by examining the dependence of SOC on RH, and the relationship between POC and carbon monoxide.

**Response:** We thank the reviewer for constructive advice. I appreciate the emphasis on thoroughly evaluating the POC and SOC results.

Firstly, we plot the concentrations of SOC against different RH levels (Figure 2) and analyze the resulting trend. It reveals that SOC levels initially rise with increasing relative humidity but subsequently decline with RH, suggesting underlying chemical or physical processes. And the result is consistent with Wang et al., (2024, Fig.4) and Zheng et al., (2015, Fig. 9b).

We also examined the relationship between POC and carbon monoxide (Figure 3). It reveals that POC is highly correlated with CO, with an $R^2$ of 0.8.

[Figure]

Figure 2 Change of SOC with RH

[Figure]

$y = 0.2035x$
$R^2 = 0.80$

Figure 3 Relationship of POC with CO

Zheng, G. J., Duan, F. K., Su, H., Ma, Y. L., Cheng, Y., Zheng, B., Zhang, Q., Huang, T., Kimoto, T., Chang, D., Pöschl, U., Cheng, Y. F., and He, K. B.: Exploring the severe winter haze in Beijing: the impact of synoptic weather, regional transport15-2969-2015, 2015.

Wang, Q., Du, W., Zhou, W., Zhang, Y., Xie, C., Zhao, J., Xu, W., Tang, G., Fu, P., Wang, Z., Sun, Y., and Peng, L.: Characteristics of sub-micron aerosols above the urban canopy in Beijing during warm seasons, Science of The Total Environment, 926, 171989, https://doi.org/10.1016/j.scitotenv.2024.171989, 2024.

**Change in the manuscript:**

We have added the evaluation of SOC and POC in the supplement file.

[Figure]

Figure S1 Evaluation of SOC formation and POC. (a) Estimation of SOC with EC-tracer method. Blue squares indicate data used to calculate primary OC/EC, while orange-filled circles indicate other OC/EC data. (b) Change of SOC with RH. (c) Relationship of POC with CO.

Second, the robustness of the measured babs, i.e., the observational constraint. For AE33, the correction factor for multiple scattering effect should be carefully determined (rather than simply using a reported value).

**Response:** Thank you very much for your valuable comments. Regarding your concern about the determination of the correction factor for the multiple scattering effect in the AE33 Aethalometer, we would like to provide further clarification and response here:

Firstly, we acknowledge the significance of the multiple-scattering correction factor in deriving absorption coefficients from the AE33 measurements, as emphasized in

various studies (Wu et al., 2024; Yus-Díez et al., 2021; Qin et al., 2018). Yus-Díez et al (2021) found that when the single-scattering albedo (SSA) of the collected particles is above a site-dependent threshold, neglecting the notable increase in the correction factor at high SSA levels can lead to a substantial overestimation of absorption coefficients obtained from Aethalometer instruments.

Secondly, regarding the observation in this study, the site-dependent experimentally multiple-scattering correction factor could not be obtained due to the lack of parallel observations. Consequently, the default multiple-scattering correction factor was utilized. As shown in the paper published before, an intercomparison of AE33 and CAPS PMssa was conducted and these results suggest that $b_{abs}$ from different measurements agree reasonably well (Xie et al., 2019, Figure 2).

Finally, a note has been included in the text stating that there exists some uncertainty in the absorption measurement due to the use of the reported value of the multiple-scattering correction factor (Yus-Díez et al., 2021).

Thank you once again for your valuable comments.

Qin, Y. M., Tan, H. B., Li, Y. J., Li, Z. J., Schurman, M. I., Liu, L., Wu, C., and Chan, C. K.: Chemical characteristics of brown carbon in atmospheric particles at a suburban site near Guangzhou, China, Atmos. Chem. Phys., 18, 16409-16418, 10.5194/acp-18-16409-2018, 2018.

Wu, L., Wu, C., Deng, T., Wu, D., Li, M., Li, Y. J., and Zhou, Z.: Field comparison of dual- and single-spot Aethalometers: equivalent black carbon, light absorption, Ångström exponent and secondary brown carbon estimations, Atmos. Meas. Tech., 17, 2917-2936, 10.5194/amt-17-2917-2024, 2024.

Xie, C. H., Xu, W. Q., Wang, J. F., Wang, Q. Q., Liu, D. T., Tang, G. Q., Chen, P., Du, W., Zhao, J., Zhang, Y. J., Zhou, W., Han, T. T., Bian, Q. Y., Li, J., Fu, P. Q., Wang, Z. F., Ge, X. L., Allan, J., Coe, H., and Sun, Y. L.: Vertical characterization of aerosol optical properties and brown carbon in winter in urban Beijing, China, Atmospheric Chemistry and Physics, 19, 165-179, 10.5194/acp-19-165-2019, 2019.

Yus-Díez, J., Bernardoni, V., Močnik, G., Alastuey, A., Ciniglia, D., Ivančič, M., Querol, X., Perez, N., Reche, C., Rigler, M., Vecchi, R., Valentini, S., and Pandolfi, M.: Determination of the multiple-scattering correction factor and its cross-sensitivity to scattering and wavelength dependence for different AE33 Aethalometer filter tapes: a multi-instrumental approach, Atmos. Meas. Tech., 14, 6335-6355, 10.5194/amt-14-6335-2021, 2021.

**Change in the manuscript:**

A new real-time loading effect compensation algorithm was adopted, which is based on a two-parallel spot measurement of optical absorption (Drinovec et al., 2015). It is worth noting that there may be some uncertainty in the absorption measurement due to the use of the reported multiple-scattering correction factor (Yus-Díez et al., 2021; Qin et al., 2018)

Please refer to Lines 108-111.

In addition, please confirm that similar to results from in-situ techniques (e.g., photoacoustic spectrometer), the AE33-based light absorption coefficients are "sensitive" to BC mixing state, e.g., by examining the relationship between babs and EC mass concentration.

**Response:** We thank the reviewer for helpful advice.

There are no parallel observations to measure absorption during the study period. However, a previous study showed that $b_{abs}$ from different measurements agree reasonably well (Xie et al., 2019). Also, long-term changes in aerosol optical properties at IAP, Beijing have been investigated by AE33 in Sun et al. (2022).

Furthermore, we examine the relationship between AE33-based light absorption coefficients ($b_{abs}$) and EC mass concentration using linear regression analysis, as shown in the figure below. The $b_{abs}$ measured by AE33 (880 nm) and EC mass concentration are highly correlated ($R^2 = 0.89$). This is consistent with the result of Wang et al. (2014).

[Figure]

Figure 4 The linear relationship between $b_{abs}$ at 880 nm from AE33 and EC mass concentration during the study period

Sun, J., Wang, Z., Zhou, W., Xie, C., Wu, C., Chen, C., Han, T., Wang, Q., Li, Z., Li, J.,

Fu, P., Wang, Z., and Sun, Y.: Measurement report: Long-term changes in black carbon and aerosol optical properties from 2012 to 2020 in Beijing, China, Atmos. Chem. Phys., 22, 561-575, 10.5194/acp-22-561-2022, 2022.

Wang, Q. Y., Huang, R. J., Cao, J. J., Han, Y. M., Wang, G. H., Li, G. H., Wang, Y. C., Dai, W. T., Zhang, R. J., and Zhou, Y. Q.: Mixing State of Black Carbon Aerosol in a Heavily Polluted Urban Area of China: Implications for Light Absorption Enhancement, Aerosol Science and Technology, 48, 689-697, 10.1080/02786826.2014.917758, 2014.

**Change in the manuscript:**

The $b_{abs}$ at 880 nm and EC mass concentration are highly correlated (Fig. S2).
Please refer to Line 235.

3、Third, based on Figure 6 and Table 2, the performance of the "CSs" scenario (i.e., core-shell mixing) appeared best for reproducing the measured light absorption coefficients. Then I could not understand why the author argued that "Partial internal mixing and partial coating are the closest to reality". For the same reason, the logic of Sections 3.3 and 3.4 was confusing.

**Response:** We thank the reviewer for the careful review of our manuscript. We are very sorry that the manuscript confused the reviewer. To clarify this point, we would like to provide explanations from the following perspectives:

Firstly, we should admit that the performance of the "CSs" scenario appeared best in terms of reproducing the measured light absorption coefficients. However, there are uncertainties associated with the calculation. A good match between simulation and observation may be caused by unphysical reasons. As shown in Fig. 3, the concentration of secondary inorganic aerosols is underestimated by the NAQPMS model while the concentration of BC is overestimated. This could probably cause an underestimation in the coating of BC. Even if the model accurately captures the physical processes of aerosols, the fraction of embedded BC (Fin) and the coating on BC could be underestimated because of the representation uncertainties in chemical formation, potentially impacting the accuracy of the absorption calculation. What's more, uncertainty exists as other factors like the morphology of the BC core and the position of the BC core inside the coating are not considered.

The measured absorption coefficient was utilized to verify the rationality of the simulation results produced by the NAQPMS+APM and FlexAOD. Building upon this foundation, sensitivity experiments were designed based on the NAQPMS and optical

module to investigate the impact of mixing state and aging processes on aerosol light absorption. As the reviewer comments before, there may exist some uncertainty in the absorption measurement in this study due to the use of the reported value of the multiple-scattering correction factor (Yus-Díez et al., 2021).

What's more, this study aims to investigate the impact of mixing state and aging processes on absorption based on the reasonable representation of the absorption coefficient.

Secondly, the CS scenario represents an idealized and simplified representation of the complex mixing state of black carbon in the atmosphere. In reality, BC particles often exist in a more complex mixing state, with varying degrees of internal mixing and coating by other aerosol components (Li et al., 2016; Reimer et al., 2019; Wang et al., 2021; Wu et al., 2021). As shown in published papers using the transmission electron microscope, only a fraction of BC is embedded in other aerosols (Li et al., 2016).

Our argument that "Partial internal mixing and partial coating are the closest to reality" is based on the recognition that the atmospheric environment is highly dynamic and heterogeneous, leading to a wide range of BC mixing states. These mixing states can vary depending on factors such as emission sources, atmospheric conditions, and transport processes. Therefore, while the "CSs" scenario may provide a good fit for certain datasets, it may not accurately represent the full range of BC mixing states observed in the atmosphere. Considering the fraction of embedded BC and secondary components coating on BC is a compromise and reasonable solution to represent the mixing state of BC in a three-dimensional model although uncertainties exist. The statement "Partial internal mixing and partial coating are the closest to reality" in Line 390 of the original manuscript is not that suitable and we have revised the description.

Finally, in Sections 3.3 and 3.4, we aimed to investigate the impact of the mixing state and aging processes on absorption and absorption enhancement. Our study provides a more comprehensive understanding of the BC mixing state by considering different scenarios, including considering the fraction of embedded BC ($F_{in}$) calculated by advanced particle microphysics module coupled in NAQPMS, considering both the mass fraction of embedded BC and secondary components coating aerosols ($F_{in}F_c$) and the detailed microphysical process (CS-APM). For example, comparing case CSs with CS-Fin, the impact of considering a fraction of embedded BC (aging process) can be investigated.

We hope that these revisions and explanations have addressed your concerns and

clarified the logic of Sections 3.3 and 3.4. Thank you once again.

**References**

Li, W. J., Sun, J. X., Xu, L., Shi, Z. B., Riemer, N., Sun, Y. L., Fu, P. Q., Zhang, J. C., Lin, Y. T., Wang, X. F., Shao, L. Y., Chen, J. M., Zhang, X. Y., Wang, Z. F., and Wang, W. X.: A conceptual framework for mixing structures in individual aerosol particles, Journal of Geophysical Research-Atmospheres, 121, 13784-13798, 10.1002/2016jd025252, 2016.

Riemer, N., P. Ault, A., West, M., L. Craig, R., and H. Curtis, J.: Aerosol Mixing State: Measurements, Modeling, and Impacts, Reviews of Geophysics, 10.1029/2018RG000615, 2019.

Wang, Y. Y., Pang, Y. E., Huang, J., Bi, L., Che, H. Z., Zhang, X. Y., and Li, W. J.: Constructing Shapes and Mixing Structures of Black Carbon Particles with Applications to Optical Calculations, Journal of Geophysical Research-Atmospheres, 126, e2021JD034620, 10.1029/2021JD034620, 2021.

Wu, Y., Xia, Y., Zhou, C., Tian, P., Tao, J., Huang, R.-J., Liu, D., Wang, X., Xia, X., Han, Z., and Zhang, R.: Effect of source variation on the size and mixing state of black carbon aerosol in urban Beijing from 2013 to 2019: Implication on light absorption, Environmental pollution, 270, 116089, https://doi.org/10.1016/j.envpol.2020.116089, 2021.

Yus-Díez, J., Bernardoni, V., Močnik, G., Alastuey, A., Ciniglia, D., Ivančič, M., Querol, X., Perez, N., Reche, C., Rigler, M., Vecchi, R., Valentini, S., and Pandolfi, M.: Determination of the multiple-scattering correction factor and its cross-sensitivity to scattering and wavelength dependence for different AE33 Aethalometer filter tapes: a multi-instrumental approach, Atmos. Meas. Tech., 14, 6335-6355, 10.5194/amt-14-6335-2021, 2021.

**Change in the manuscript:**

Among the three ideal mixing state assumptions in this study, the result of the core-shell mixing state assumption is closest to the observed absorption coefficient. It should be noted that there are uncertainties associated with the calculation. As shown in Fig. 3, the concentration of secondary inorganic aerosols is underestimated by the NAQPMS model while BC is overestimated, therefore, the coated thickness can be underestimated. Please refer to Lines 267-271.

Considering the fraction of embedded BC and secondary components coating on BC is a compromise and reasonable solution to represent the mixing state of BC in a threedimensional model although uncertainties exist.

Please refer to Lines 398-399.

---

## Author Comment (AC2)

Response to comments from reviewers on "Modeling simulation of aerosol light absorption over the Beijing-Tianjin-Hebei region: the impact of mixing state and aging process" by Huiyun Du et al.

We thank the reviewers for their valuable comments. We have revised the manuscript according to the suggestions and responded to their concerns below (in blue).

This study uses the APM model combined with observations to discuss the impact of representative schemes of aerosols on optics. The whole study is meaningful and helpful for the experiment and model development. However, excessive use of concepts to represent aerosol mixing states lacks detailed and intuitive introductions, which reduces readability. A minor revision should be added before accepting.

1.  Many excellent concept maps can be referenced to enhance readers' understanding of mixing states, such as Fig. 4 in 10.1038/s41467-018-05635-1, Fig. 1 in 10.1175/bams-d-16-0028.1

Response: Thank you for your constructive suggestion and for highlighting the valuable resources. Incorporating high-quality concept maps into our manuscript is important to enhance readers' comprehension of mixing states.

We reviewed the concept maps from the cited articles (Fig. 4 in 10.1038/s41467-018-05635-1 and Fig. 1 in 10.1175/bams-d-16-0028.1). Matsui et al. (2018, Fig. 4) showed the impact of resolving the mixing state on the direct radiative effect of black carbon. Fierce et al. (2017, Fig.1) showed the complex particle-resolved and reduced presentation of the mixing state.

We have added the references in the Introduction section to make it easy for the reader to understand the complex concepts. Also, we have added an abstract figure to the manuscript to clarify the mixing state considered in this study. Thank you again for your comments.

**Changes in the manuscript:**

The concept maps illustrating the mixing state can be found in Matsui et al. (2018, Fig. 4) and Fierce et al. (2017, Fig.1).

Matsui, H., Hamilton, D. S., and Mahowald, N. M.: Black carbon radiative effects highly sensitive to emitted particle size when resolving mixing-state diversity, Nature Communications, 9, 3446, 10.1038/s41467-018-05635-1, 2018.

Fierce, L., Riemer, N., and Bond, T. C.: Toward Reduced Representation of Mixing State for Simulating Aerosol Effects on Climate, Bulletin of the American Meteorological Society, 98, 971-980, 10.1175/bams-d-16-0028.1, 2017.

[Figure]

Abstract figure

2. Line 40: add references for condensation and coagulation processes: 10.1016/j.isci.2023.108125

**Response:** Thank you for your suggestion. We have added the reference in the manuscript.

Chen, X., Ye, C., Wang, Y., Wu, Z., Zhu, T., Zhang, F., Ding, X., Shi, Z., Zheng, Z., and Li, W.: Quantifying evolution of soot mixing state from transboundary transport of biomass burning emissions, iScience, 26, 108125, 10.1016/j.isci.2023.108125, 2023.

**Changes in the manuscript:**

The aerosol mixing state is dynamic and changes due to several processes, such as emission, new particle formation, transport, condensation, and coagulation processes (Chen et al., 2023). Please refer to Lines 43, 469.

3. **Lines 190-192:** Fin and Fc are not clear? Number fraction? Mass fraction?

**Response:** We are very sorry to make the reviewer confused. Fin and Fc are both mass fractions. Fin means the mass fraction of embedded BC. Fc is the mass fraction of coating aerosols (the secondary aerosols coated on BC). We have changed the description in the revised manuscript.

**Changes in the manuscript:**

Secondly, to see the impact of the aging process, simulations were designed using a partial core-shell mixing state in FlexAOD, including CS-Fin (Fin mass fraction of embedded BC core) and CS-FinFc (Fc mass fraction of coating aerosols coating on Fin mass fraction of embedded BC) (Fig. S3). Please refer to Lines 194-196.

4. **What are the differences between CS-Fin and CS-FinFc? You divided accumulation mode aerosols into 4 types (embedded, partly coated, bare-like BC and BC-free) or 3 types (embedded, bare-like BC and BC-free)? Detailed**

**introductions should be added for mixing states in Table 1.**

**Response:** Thank you for your constructive suggestion.

The differences between CS-$F_{in}$ and CS-$F_{in}F_c$ lie in whether the fraction of secondary aerosols coated on BC is considered.

CS-$F_{in}$ refers to a scenario where the $F_{in}$ mass fraction of BC aerosols is embedded, with all secondary aerosols coating on embedded BC (as illustrated in the third panel of Figure below).

CS-$F_{in}F_c$ refers to a scenario where the $F_{in}$ mass fraction of BC is embedded, and the $F_c$ mass fraction of secondary aerosols is coated on embedded BC (as illustrated in the fourth panel of Figure below).

In this study, aerosols can be divided into three types (embedded, bare-like BC, and BC-free) under the CS-FinFc scenario. Furthermore, detailed introductions will be added to Table 1 in the manuscript.

[Figure]

Figure S3 The concept of different mixing state assumption

**Changes in the manuscript:**

**Table 1 Simulation test design**

| Case | Method | Input | Size distribution | Mixing state |
|------|--------|-------|-------------------|--------------|
| EXT$_O$ | FlexAOD | observed | fixed | external |
| HOM$_O$ | FlexAOD | observed | fixed | internal homogeneous |
| CS$_O$ | FlexAOD | observed | fixed | core-shell |
| EXT$_S$ | FlexAOD | simulated | fixed | external |
| HOM$_S$ | FlexAOD | simulated | fixed | internal homogeneous |

| | | | | |
|---|---|---|---|---|
| CS$_S$ | FlexAOD | simulated | fixed | core-shell |
| CS-F$_{in}$ | FlexAOD | simulated | fixed | partial core-shell and partial bare BC [a] |
| CS-F$_{in}$F$_c$ | FlexAOD | simulated | fixed | partial core-shell, partial bare BC and partial coating aerosols [b] |
| CS-APM | APM | simulated | simulated | semi-external (hourly) [c] |

| Impact | Description |
|---|---|
| EXT$_O$ vs. HOMo vs. CSo | Impact of mixing state when inputting observed data |
| EXT$_S$ vs. HOM$_S$ vs. CS$_S$ | Impact of mixing state when inputting simulated data |
| CSo vs. CSs | Impact of aerosol mass concentration |
| CSs vs. CS-F$_{in}$ | Impact of aging process (fraction of embedded BC) |
| CSs vs. CS-F$_{in}$F$_c$ | Impact of the aging process (fraction of embedded BC and coating shell) |
| CS-F$_{in}$F$_c$ vs. CS-APM | Impact of detailed microphysical process |

[a] Aerosols are divided into two types: embedded, bare-like BC aerosols.

[b] Aerosols are divided into three types: embedded, bare-like BC, and BC-free aerosols.

[c] Concept map can be referred to Chen et al. (2019, Fig1)

Chen, X., Yang, W., Wang, Z., Li, J., Hu, M., An, J., Wu, Q., Wang, Z., Chen, H., Wei, Y., Du, H., and Wang, D.: Improving new particle formation simulation by coupling a volatility-basis set (VBS) organic aerosol module in NAQPMS+APM, Atmospheric Environment, 204, 1-11, 10.1016/j.atmosenv.2019.01.053, 2019.

Please refer to Lines 199-202, Lines 466-468.

**5. How to define Partial internal mixing and partial coating?**

**Response:** We apologize for this confusion caused by our terminology.

In this study, partial internal mixing and partial coating have the same meaning as CS-F$_{in}$F$_c$. Partial internal mixing means only part of the black carbon particles is core-shell

mixed with the secondary component, and partial coating means only part of the secondary aerosols are coated on BC.

We appreciate your attention to detail. To avoid redundancy, we will uniformly adopt CS-$F_{in}F_c$ throughout the manuscript and omit unnecessary repetitions.

**Changes in the manuscript:**

We rewrite the sentences "Partial internal mixing and partial coating" to avoid redundancy and confusion.

"Considering the fraction of embedded BC and secondary components coating on BC is a compromise and reasonable solution to represent the mixing state of BC in a three-dimensional model although uncertainties exist".

Please refer to Line 398.

**6. Line 313: How do you calculate Eabs? Add detailed calculation/inversion process**

**Response:** Thank you for your constructive suggestion. The calculation of $E_{abs}$ and the impact of the mixing state on Eabs were investigated in 3.4. We have added the detailed calculation process to this part.

In this study, the BC absorption enhancement is the ratio of the absorption coefficient calculated assuming core-shell mixing to that calculated using external mixing.

$$E_{abs} = \frac{b_{abs}(\lambda, coreshell\ mixing)}{b_{abs}(\lambda, external\ mixing)}$$

Therefore, the absorption enhancements in core-shell mixing, CS-Fin, and CS-FinFc cases are the ratio of absorption under those cases to absorption under external mixing. In the CS-APM case, as described in Line 330-334, the radiative absorption enhancement is the ratio of the absorption coefficient in the base simulation to that in the sensitivity test turning off the coating process.

**Changes in the manuscript:**

The BC absorption enhancement is calculated as the ratio of the absorption coefficient calculated assuming core-shell mixing to that calculated using external mixing.

$$E_{abs} = \frac{b_{abs}(\lambda, coreshell\ mixing)}{b_{abs}(\lambda, external\ mixing)}$$

Please refer to Lines 328-331.

---

## Author Response (AR2)

Response to comments from reviewers on "Modeling simulation of aerosol light absorption over the Beijing-Tianjin-Hebei region: the impact of mixing state and aging process" by Huiyun Du et al.

We thank the reviewers for their valuable comments and constructive suggestions. We have revised the manuscript according to the suggestions and responded to their concerns below (in blue).

Reviewer #1

I am generally satisfied with the responses and the revised manuscript. After clarifying the following minor concerns, it should be publishable. Page numbers mentioned below are for the PDF file "Reply to RC1".

1. Why did the POC vs. CO plots shown in Pages 5 and 6 look different?

**Response:** Sorry to make the reviewer confused. The two figures are based on the same dataset, but the x-axis and y-axis in two figures are reversed. Specifically:

The figure in Page 6 shows POC vs. CO, while the figure in Page 5 shows CO vs. POC. For consistency, we have retained the POC vs. CO plots in the supplement file.

What's more, we have corrected the unit error in the FigureS1b.

**Change in the manuscript:**

Please refer to FigureS1 in the supplement file.

[Figure]

Figure S1 Evaluation of SOC formation and POC. (a) Estimation of SOC with EC-tracer method. Blue squares indicate data used to calculate primary OC/EC, while orange-filled circles indicate other OC/EC data. (b) Change in SOC with RH. (c)

Relationship between POC and CO.

2. Page 6, Figure S1 (b), suggest investigating the dependence of SOC to EC ratio on RH, and checking whether the related statements still held.

**Response:** We examined the dependence of SOC to EC ratio on RH (Figure 1a). It reveals that SOC/EC levels also initially rise with increasing relative humidity and subsequently decline with RH. And there exists another little peak of SOC/EC when RH is about 62%.

[Figure]

Figure 1 Dependence of (a) SOC to EC ratio and (b) SOC on RH